# Effects of acupuncture on earthquake survivors with major psychiatric disorders and related symptoms: A scoping review of clinical studies

**Chan-Young Kwon**[1☯]**, Jungtae Leem**[2☯]**, Da-Woon Kim**[3]**, Hui-Ju Kwon**[4]**, Hyun-Seo Park**[5]**, Sang-Ho Kim**[3] *

**1** Department of Oriental Neuropsychiatry, Dong-Eui University College of Korean Medicine, Busan, Republic of Korea, **2** College of Korean Medicine, Wonkwang University, Iksan, Republic of Korea, **3** Department of Neuropsychiatry of Korean Medicine, Pohang Korean Medicine Hospital Affiliated to Daegu Haany University, Pohang-si, Gyeongsangbuk-do, Republic of Korea, **4** College of Korean Medicine, Daegu Haany University, Gyeongsan-si, Gyeongsangbuk-do, Republic of Korea, **5** Department of Internal Medicine of Korean Medicine, National Medical Center, Seoul, Republic of Korea

☯ These authors contributed equally to this work.
* omed22@naver.com

**Data Availability Statement:** All relevant data are within the paper and its Supporting information files.

## Abstract

### Background

This scoping review aimed to determine the current research status of acupuncture for major psychiatric disorder (MPD) in earthquake survivors.

### Method

We followed the scoping review process described previously. A literature search on 14 electronic databases was conducted from inception to November 29, 2022. Data from the included studies were collected and descriptively analyzed to address our research question. Extracted data were collated, synthesized, and summarized the according to the analytical framework of a scoping review.

### Result

This scoping review included nine clinical studies: four randomized controlled trials (RCTs) and five before–after studies. The most frequent MPD type among the included acupuncture studies was posttraumatic stress disorder (PTSD; 6/9, 66.67%). The most frequent acupuncture type was scalp electro-acupuncture (4/9, 44.44%), followed by manual acupuncture and ear acupressure/ear acupuncture (3/9, 33.33%). Studies using scalp electro-acupuncture all used common acupoints, including GB20, GV20, GV24, and EX-HN1. In general, the treatment period lasted between 4 and 12 weeks. Validated assessment tools for PTSD severity and accompanying symptoms were used for patients with PTSD, while the corresponding evaluation tools were used for patients with other diagnoses or clinical symptoms. Acupuncture-related adverse events were generally mild and temporary, such

**Funding:** This work was supported by the National Research Foundation of Korea (NRF) grant funded by the Korean government (MSIT) (No. 2021R1F1A105928211). The funding source had no input in the interpretation or publication of the study results (※MSIT: Ministry of Science and ICT). The funders had no role in study design, data collection and analysis, decision to publish, or preparation of the manuscript.

**Competing interests:** The authors have declared that no competing interests exist.

as mild bleeding and hematoma, and syncope was a rare but potentially serious adverse event (1/48 patients and 1/864 sessions over a treatment period of 4 weeks).

## Conclusion

Acupuncture studies for MPD after an earthquake mainly focused on PTSD. RCTs accounted for around half of the included studies. Scalp electro-acupuncture was the most common acupuncture type, and EX-HN1 and GV24 were the most important acupoints in the acupuncture procedures for MPD. The included studies mostly used validated symptom assessment tools, though some did not. Clinical studies in this field need to be further expanded regardless of the study type.

## Protocol registration

https://osf.io/wfru7/.

## Introduction

Among all natural disasters, earthquakes have been considered to have the greatest destructive effects given that they cause not only physical impairments but also psychological stresses among victims [1]. Over the past several decades, earthquakes have drawn attention due to their frequent occurrence and massive destruction [2]. In fact, the past decade has seen around 1,300 to more than 2,500 magnitude of 5.0 or higher earthquakes every year worldwide [3]. Given that natural disasters, including earthquakes, are usually unpredictable and destructive, some survivors suffer severe trauma-related symptoms [4]. Disaster survivors with high levels of exposure to earthquakes have a 1.4 times higher prevalence of major psychiatric disorder (MPD), such as posttraumatic stress disorder (PTSD), major depressive disorder (MDD), other anxiety disorders, and nicotine dependence, than do non-exposed survivors [4].

Typically, pharmacological and psychological treatments have been currently employed for MPD following disasters [5–7]. In line with this, selective serotonin reuptake inhibitors have typically been used as a pharmacological treatment for MPD. However, this approach has been associated with adverse effects, such as constipation, diarrhea, dizziness, nausea and sexual dysfunction [6, 8, 9]. Although benzodiazepines have also been used for MPD, they have been shown to cause drowsiness, falls, or overdoses and even carry significant risk for medical complications, including delirium [10]. Psychological treatments include eye movement desensitization therapy, prolonged exposure, cognitive processing therapy, and cognitive behavioral therapy (CBT) [11]. Psychological treatment requiring an expert practitioner is not always feasible, with long treatment waiting lists resulting from a limited number of qualified therapists [12].

Acupuncture is a type of complementary integrative medicine that may be considered for the treatment of MPD [13]. A number of studies regarding the effectiveness and safety of acupuncture as a treatment of PTSD, MDD, and anxiety disorder have been published [14–16]. Moreover, several previous studies have used acupuncture for the treatment of PTSD, MDD, and anxiety disorder among earthquake survivors [17, 18]. Acupuncture can be used as an immediate medical treatment when medical resources are scarce, with evidence suggesting its effectiveness for not only physical but also psychological symptoms [19]. Moreover, patients with PTSD, who often experience chronic pain [20], may benefit considerably from

acupuncture given its ability to ease pain [21]. Acupuncture may be an easier and cheaper treatment approach than CBT or pharmacological treatment [22].

Evidence suggests that a scoping review is more appropriate than a systematic review when exploring a wide range of questions related to, for example, the type of research design used, the concepts and characteristics of existing literature, or the identification knowledge gaps [23]. Given the absence of active acupuncture research on PTSD among earthquake survivors, our research team sought to conduct a scoping review that would provide a wider view of the relevant field. This scoping review aimed to identify which type of clinical research design had been utilized to study acupuncture treatment for MPD in earthquake survivors. Specifically, the authors focused on detailed methodological issues, such as treatment regimens, detailed characteristics of the participants, and frequently used outcomes.

## Methods

### Study design and registration

This scoping review followed the method developed by Arksey, O'Malley [24], and others [25, 26], as well as the Preferred Reporting Items for Systematic reviews and Meta-Analyses Extension for Scoping Reviews guidelines [26]. The protocol for this study was submitted to the Open Science Framework on July 19, 2022, which was later updated on November 29, 2022, and can be accessed at https://osf.io/wfru7/. The protocol paper for this scoping review was published in January 2023 [27]. The detailed research methods of this study are described in the protocol paper [27]. Given that this study was based exclusively on published literature, ethics approval and informed consent were not required.

### Stage 1: Identifying the study questions

During stage 1 of the scoping review, the research team searched related articles collaboratively to identify the most appropriate research questions. The team, which consisted of three specialists in neuropsychiatry (SHK, CYK, and DWK), a specialist in clinical research on traditional East Asian medicine (JTL), and an undergraduate researcher (HJK), used existing clinical evidence from the literature to develop a set of questions for the review. The team reached a consensus on the final set of research questions after multiple rounds of revisions and discussions. The following questions addressed a range of topics related to the use of acupuncture for the treatment of MPD in earthquake survivors: [1] *which clinical research designs were used in previous studies?*, [2] *which populations were targeted in previous acupuncture studies?*, [3] *what were the most frequently used acupuncture types for MPD management?*, [4] *what was the appropriate length of acupuncture treatment for MPD management?*, [5] *what clinical outcomes were adopted in previous studies?*, and [6] *what type of adverse events occurred after acupuncture therapy?* The research team generated these questions through consensus and agreement on revisions.

### Stage 2: Identifying relevant studies

**Information source.** To identify relevant studies for the scoping review, a comprehensive literature search was conducted on 14 electronic databases, including Medline (via PubMed), Excerpta Medica dataBASE, Cochrane Central Register of Controlled Trials, Allied and Complementary Medicine Database, Cumulative Index to Nursing and Allied Health Literature, PsycArticles, SCOPUS, Web of Science, China National Knowledge Infrastructure, Wanfang, VIP, Oriental Medicine Advanced Searching Integrated System, Korea Citation Index, and Citation Information by NII (initial search: 2022.07.13., updated search: 2022.11.29.). The

search strategy was developed after consulting with a clinical researcher, a literature review expert, and a specialist on psychiatric diseases. The search terms comprised a combination of exposure (earthquake) and intervention (acupuncture) terms, along with various synonyms and related medical subject headings. In terms of exposure, the search was not restricted to a particular type of MPD, and only terms associated with earthquakes were used in the search phrase (S1 Appendix).

**Eligibility criteria.** (1) Study design: The study designs of the publications included in this scoping review were restricted to randomized/quasi-randomized/non-randomized controlled clinical trials, single-arm trials, case series, cross-sectional studies, and feasibility studies. However, case reports with fewer than three patients [28], literature reviews, and preclinical studies were excluded. There were no restrictions on the publication language of the study. (2) Participant type: Regarding participant type, this review included earthquake event survivors with PTSD, MDD, and anxiety disorders, including related symptoms. No restrictions were placed on the diagnostic criteria for MPD, such as the Diagnostic and Statistical Manual of Mental Disorders and International Classification of Diseases. (3) Intervention type: Except for acupressure, various acupuncture therapies were allowed, including manual acupuncture, electro-acupuncture, warm-needle acupuncture, fire-needle acupuncture, bee-venom acupuncture, pharmacopuncture, and acupotomy. No restrictions were placed on the treatment period, dosage, treatment frequency, and concomitant treatment. (4) Control group intervention: Any type of intervention was allowed for the control group, except for traditional East Asian medicine interventions, such as herbal medicine, moxibustion, cupping, and tuina. (5) Outcome: The outcome measures included all symptoms related to the diagnosis of MPD, adverse events, and dropout rates related to treatment. PTSD outcomes were categorized into three groups according to a previous study [29], with outcomes related to (1) the psychological aspect (e.g., anxiety, fear, anger, irritability, guilt, shame, apathy, distrust, sadness, frustration, alienation, loss of confidence, and mourning), (2) the somatic aspect (e.g., insomnia, palpitation, pain, anorexia, and fatigue), and (3) cognitive aspects (e.g., decreased memory, difficulty making decisions, repeated recall of traumatic events, and difficulty concentrating).

## Stage 3: Study selection

In the third stage of the study, two reviewers (HJK and DWK) independently removed duplicate publications and then assessed the titles and abstracts of the articles based on the inclusion criteria. In the next phase, the full texts of the articles were also assessed for eligibility. Any discrepancies were resolved through discussions with a third independent researcher (SHK). The detailed process of study selection is illustrated in Fig 1.

## Stage 4: Charting the data

A data extraction sheet was developed by the research team after conducting a pilot test. The extraction sheet consisted of several items, including (1) general information (e.g., author's name, country, publication year, and research design); (2) participant demographic data (e.g., age, sex, number of participants, diagnostic criteria, disease duration and severity, and initial and final number of participants); (3) intervention information (e.g., type of acupuncture, acupuncture points, treatment dosage, treatment period, and control/concomitant intervention information); and (4) outcome variables including effects, safety, and research findings. The data extraction was performed independently by two reviewers (HJK and DWK) and then crosschecked. Any discrepancies were resolved through discussions with a third independent researcher (SHK).

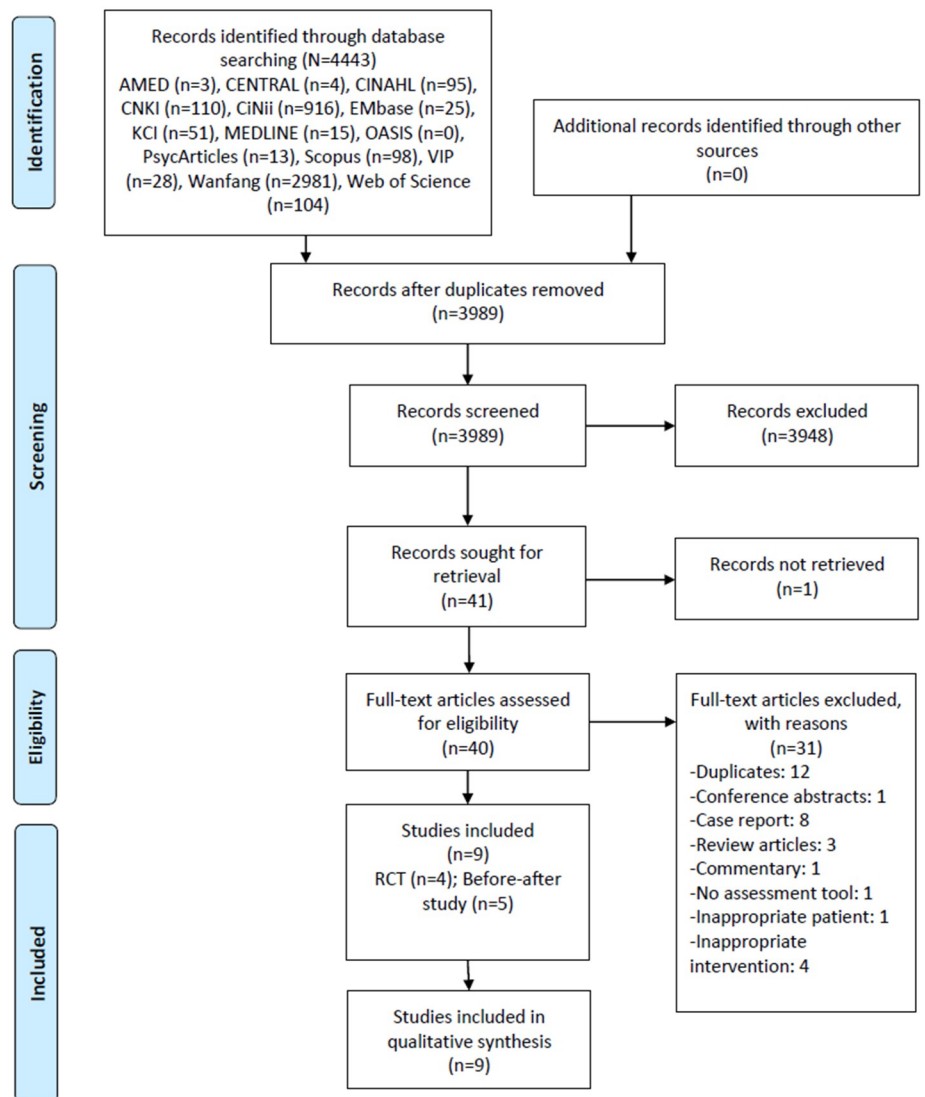

**Fig 1. PRISMA flow chart for the study selection process.**

## Stage 5: Collating, summarizing, and reporting the results

The data extracted from the included studies were synthesized and summarized using the scoping review analytical framework. In the qualitative analysis stage, the characteristics of the included studies were described, such as the author's name, country, publication year, earthquake details, number of participants, sex, age, research design, and type of treatment/control interventions. A second table titled "Detailed information of acupuncture treatment" was created to summarize the details of the acupuncture intervention, such as acupuncture type, location of acupuncture points, number of treatments, treatment frequency, treatment period, and details of control/concomitant interventions. The Standards for Reporting Interventions in Clinical Trials of Acupuncture reporting guideline [30] was used for the detailed analysis of the acupuncture regimen used. The third table, titled "Effects and safety of acupuncture treatment for MPD and related symptoms," included the outcomes, adverse events, and conclusions of each study. Finally, a research map table was provided to help with future research

planning by allowing the visualization of the state of ongoing research and commonly utilized findings.

In addition, the methodological quality of the studies included in this review was assessed. Corresponding assessment tools were used to assess the methodological quality depending on study type. As the studies included in this review were RCTs and before–after studies, Cochrane's risk of bias tool [31] and the Quality Assessment Tool for Before–After (Pre–Post) Studies With No Control Group developed by the National Heart, Lung, and Blood [32], respectively, were used. The methodological quality of the included studies was assessed by two independent researchers (DWK and SHK), and in the case of a disagreement during the evaluation process, both of them would discuss it and arrive at a consensus.

## Results

### Literature search and study selection

In the initial search, 3989 records were obtained, excluding duplicate documents. In the first screening process, 41 potentially relevant records were screened, among which 40 underwent full text reviews, except for one record whose full text was unavailable. In the second screening process, 12 duplicates, 1 conference abstract, 8 case reports, 3 review articles, 1 commentary, 1 without a description of the assessment tool, 1 with inappropriate patients, and 4 with inappropriate interventions were excluded. Ultimately, a total of nine clinical studies [17, 33–40] were included in this scoping review (Fig 1).

### General characteristics of included studies

Among the included studies, four [17, 33–35] were randomized controlled trials (RCTs), whereas other five [36–40] were before–after studies. All studies had been published from 2009 to 2020.

**Randomized controlled trials.** (1) Population: Among the four [17, 33–35] included RCTs, three [17, 33, 34] were related to the Wenchuan earthquake and one [35] was related to the Dingxi earthquake. Among the four [17, 33–35] studies, two [33, 34] involved patients with PTSD, one [17] involved patients with generalized anxiety disorder (GAD), and one [35] involved patients with depression. (2) Intervention: Four acupuncture types were used, including scalp electro-acupuncture in two studies [33, 34], manual acupuncture in two studies [17, 35], ear acupressure in one study [33], and hydro-acupuncture in one study [17]. Two studies used moxibustion as a combination treatment with acupuncture [33, 34]. The duration of treatment was four weeks (or 1 month) in two studies [17, 35] and 12 weeks in the other two studies [33, 34] (3) Control: Most interventions in the control group were pharmacotherapies, including paroxetine and buspirone, with one study [35] providing psychotherapy as a control intervention. (4) Outcome: In both studies involving patients with PTSD [33, 34], the Clinician-Administered PTSD Scale (CAPS) was used to assess the severity of PTSD symptoms, whereas the Hamilton Anxiety Rating Scale (HAMA) and the Hamilton Depression Rating Scale (HAMD) were used to assess anxiety and depression, respectively. The study involving patients with GAD [17] evaluated anxiety using the HAMA and the Zung Self-Rating Anxiety Scale (SAS), whereas the other study involving patients with depression [35] evaluated depression using the HAMD. Moreover, both of these studies [17, 35] used the total effective rate (TER), which evaluates the improvement in overall clinical symptoms, as their outcome (Table 1).

**Before–after studies.** (1) Population: Among the five included before–after studies [36–40], three [36–38] were related to the Wenchuan earthquake, one [39] to the Central Italy earthquake, and one [40] to the Pohang earthquake. Four [36–38, 40] of the five studies

**Table 1. Characteristics of included randomized controlled trials.**

| Study (country) | Sample size (included→analyzed) | Mean age (range) (years) | Sex (Male: Female) | Population (diagnostic tool) | Related disaster (magnitude/date) | (A) Treatment intervention | (B) Control intervention | Duration of treatment/f/u | Outcome | Results reported | Adverse events reported |
|---|---|---|---|---|---|---|---|---|---|---|---|
| Zhang2010-a (China) [33] | 276 (69:69:69:69) → 250 (65:62:65:64) | 18–65 | 94:162 | PTSD (DSM-IV) | Wenchuan earthquake (8.0/May 2008) | (a) Scalp electro-acupuncture (b) Scalp electro-acupuncture with moxibustion (c) Scalp electro-acupuncture with ear acupressure | (d) Paroxetine 20 mg/d | 12 weeks/6 months | (1) CAPS (2) CAPS reduction ratio (3) HAMD (4) HAMD reduction ratio (5) HAMA (6) HAMA reduction ratio | (1-a, d) N.S (1-b, d) N.S (1-c, d) N.S (2-a, d) (a)>(d)[+] (2-b, d) N.S (2-c, d) N.S (3-a, d) N.S (3-b, d) N.S (3-c, d) N.S (4-a, d) (a)>(d)[+] (4-b, d) N.S (4-c, d) N.S (5-a, d) N.S (5-b, d) N.S (5-c, d) N.S (6-a, d) (a)>(d)[+] (6-b, d) (b)>(d)[+] (6-c, d) N.S | (A): mild bleeding, hematoma, pain, and syncope after acupuncture (B): leukopenia 1 and other adverse events in behavior, autonomic nervous, cardiovascular, and gastrointestinal systems. |
| Zhang 2010-b (China) [17] | 325(186:139) | (A): 45.69 ± 5.54 (12–58) (B): 46.90 ± 5.89 (12–60) | 144:181 | Generalized anxiety disorder (CCDM-3) | Wenchuan earthquake (8.0/May 2008) | Acupuncture + hydro-acupuncture | Buspirone 15 mg/d | 4 weeks | (1) HAMA (2) SAS (3) TER* (3–1) Total cure and marked effective rate | (1) N.S (2) N.S (3) N.S (3–1) (A)>(B)[+] | (A): acupuncture syncope 3 (B): dizziness 15, dry mouth 12, headache 4, nausea 2, constipation 8, insomnia 3, and dysuria 3 |

*(Continued)*

**Table 1.** (Continued)

| Study (country) | Sample size (included→analyzed) | Sex (Male: Female) | Mean age (range) (years) | Population (diagnostic tool) | Related disaster (magnitude/ date) | (A) Treatment intervention | (B) Control intervention | Duration of treatment/ f/u | Outcome | Results reported | Adverse events reported |
|---|---|---|---|---|---|---|---|---|---|---|---|
| Zhang 2010-c (China) [34] | 92(46:46) →81(41:40) | 32:49 | (A): 47.41 (24–65) (B): 47.98 (18–65) | PTSD (DSM-IV) | Wenchuan earthquake (8.0/May 2008) | Scalp electro-acupuncture + moxibustion | Paroxetine 20 mg/d | 12 weeks | (1) CAPS severity (2) CAPS frequency (3) CAPS severity reduction ratio (4) CAPS frequency reduction ratio (5) HAMD (6) HAMD reduction ratio (7) HAMA (4) HAMA reduction ratio | (1) (A)< (B)+ (2) (A)< (B)+ after 6 weeks, N.S after 12 weeks (3) (A)> (B)+ (4) (A)> (B)+ after 6 weeks, N.S after 12 weeks (5) (A)< (B)+ after 12 weeks N.S after 6 weeks, (6) (A)> (B)+ after 6 weeks, N.S after 12weeks (7) N.S after 6 weeks, (A)<(B)+ after 12 weeks (8) N.S after 6 weeks, (A)>(B)+ after 12 weeks | NR |
| Zhao 2014 (China) [35] | 90(30:30:30) | 51:39 | 10–75 | Depression (CCMD-3) | Dingxi earthquake (6.0/July 2013) | (a) Warming-promotion acupuncture (b) Regular acupuncture | (c) Psychotherapy | 1 month | (1) TER* (1–1) Total cure rate (2) HAMD | (1) N.S (1–1) (a)>(b)++, (a)>(c)++ (2) (a)< (b)+, (a)< (c)++ | NR |

*The total effective rate is divided into cure, marked effective, effective, and failure.

"+" and "++" indicate significant differences between two groups; p < 0.05 and p < 0.01, respectively. "N.S" means no significant difference between the two groups; p > 0.05.

Abbreviations: CAPS, Clinician-Administered PTSD Scale; CCMD, Chinese Classification of Mental Disorders; DSM, Diagnostic and Statistical Manual of Mental Disorders; HAMA, Hamilton Anxiety Rating Scale; HAMD, Hamilton Depression Rating Scale; PTSD, Post-Traumatic Stress Disorder; SAS, Zung Self-Rating Anxiety Scale; TER, Total effective rate.

involved patients with PTSD, whereas one [39] involved patients with musculoskeletal pain and psychologic symptoms. (2) Intervention: Except for one study [39] that did not describe the acupoint stimulated, three acupuncture types were used in the four before-after studies [36–38, 40], including scalp electro-acupuncture in two studies [36, 38], ear acupressure or ear acupuncture in two studies [36, 40], and manual acupuncture in one study [37]. Two studies [36, 39] used moxibustion as a combination treatment with acupuncture. In one study [39], the treatment period was 4 days, whereas in the other studies [36–38, 40], the treatment period lasted 8–12 weeks. (3) Outcome: Among the four studies [36–38, 40] involving patients with PTSD, only two [38, 40] used validated assessment tools for PTSD severity, including the CAPS and Impact Event Scale-Revised Korean Version (IES-R-K). In addition, both of these studies [38, 40] used scales to evaluate common symptoms in patients with PTSD, such as anxiety (HAMA), depression [HAMD and Patient Health Questionnaire-9 (PHQ-9)], anger [State-Trait Anger Expression Inventory (STAXI)], and sleep disturbance [Pittsburgh Sleep Quality Index (PSQI)]. Moreover, Kim et al. (2020) [40] assessed the quality of life of patients with PTSD using EuroQol 5-Dimension (EQ-5D). In the other two studies [36, 37] involving patients with PTSD, only TER was used as the outcome. In one study [39] involving patients with musculoskeletal pain and psychologic symptoms, the verbal numerical scale was used to assess the severity of heterogeneous symptoms (Table 1).

## Details regarding the acupuncture methods for MPD

The acupuncture methods used in the included studies can be classified into the following four categories.

**Scalp electro-acupuncture (n = 4).** All four studies [33, 34, 36, 38] using scalp electro-acupuncture shared the same acupoints, including GB20, GV20, GV24, and EX-HN1. All acupuncture procedures included the *de qi* procedure. The method for electronic stimulation in the four studies [33, 34, 36, 38] was a continuous wave. Three studies [33, 36, 38] used a frequency of 100 Hz, whereas the other study [34] used a frequency of 300–500 times/min (approximately 5–8 Hz). A total of 36 treatment sessions were performed in three studies [33, 36, 38], whereas 18 sessions were performed in one study [34]. In the four studies [33, 34, 36, 38]. the treatment frequency was three sessions per week. The retention time was 30 min in three studies [33, 34, 38] and 20 min in one study [36].

**Manual acupuncture (n = 3).** Although the three studies [17, 35, 37] that performed manual acupuncture used heterogeneous acupoints, all three studies shared a common acupoint, namely EX-HN1. Although one study [37] did not report on the achievement of *de qi*, the other two [17, 35] included the *de qi* procedure in the acupuncture procedure. A total of 20–30 acupuncture sessions were conducted in these studies [17, 35, 37]. In principle, the frequency of treatment was daily; however, discrepancies were observed with regard to whether participants took 1 day off every week or 2 days off every 10 days. Except for one study [17], which did not report the retention time, the remaining two studies [35, 37] had a retention time of 30 min.

**Ear acupressure or ear acupuncture (n = 3).** A total of seven ear acupoints were used in the three studies [33, 36, 40], all of which used the subcortex, sympathetic, liver, and kidney acupoints. In two studies [33, 36], ear acupressure was performed using Wang-Bu-Liu-Xing seeds, whereas in the other study [40], it was performed using press needles. In all three studies [33, 36, 40], unilateral ear acupoints were stimulated at each session; however, the attachment of the seeds or needles was changed to the contralateral ear acupoints thrice or twice a week. One study [40] stated that the participants were recommended to self-stimulate the attached needles.

**Hydro-acupuncture (n = 1).**   After disinfection at three acupoints, namely *minghuang*, *tianhuang*, and *qihuang*, 1 mL of medication was injected to stimulate the acupoints (Table 2) [17].

## Methodological qualities of included studies

Among the four RCTs [17, 33–35] included, two [17, 33] were considered to use random sequence generation with a low risk of bias (e.g., computerized random number generation). In one study [34], the random sequence generation method was not described. In only one study [33], allocation concealment was performed and described. A study [35] in which participants were assigned to treatment or control groups in the order of treatment was evaluated as high risk in this domain. All studies [17, 33–35] were rated as unfavorable in terms of double-blindness because it was judged that double-blindness would not be possible because of the study design. There was only one study [33] reporting blinding of outcome assessments. One study [34] was evaluated as having a high risk of bias in the incomplete outcome data domain because the participant dropout rate was high (approximately 12%) and intention-to-treat analysis was not performed.

In all five included before–after studies [36–40], the study purpose was clearly stated. However, only one study [38] clearly stated inclusion and exclusion criteria of participants. Except for one study [39], the interventions used were clearly described in the remaining studies. Although the study [39] used different types of interventions depending on the patients included, it did not provide detailed descriptions of those interventions. Two studies [38, 40] clearly described the outcome used. Because TER is not generally considered a validated and definitive outcome, two studies [36, 37] using only this outcome were rated "No" for this question. One study using only the verbal/numerical scale [39] was also rated "No" for this question. Only one study [38] reported blinding of outcome assessment. Follow-up was conducted in one study [40]; however, the rate of loss to follow-up was 25%. In three studies [38–40], appropriate statistical analysis was performed for before–after comparisons, p-values were presented, and multiple-time outcome indicators were evaluated after the intervention (S2 Appendix).

## Reported effectiveness and safety of acupuncture for MPD

**Effectiveness in randomized controlled trials.**   The two RCTs [33, 34] involving patients with PTSD showed that 12 weeks of scalp electro-acupuncture or a combination of scalp electro-acupuncture and moxibustion promoted a significantly greater reduction in the CAPS score compared to 20 mg of paroxetine per day. Moreover, both studies [33, 34] found that scalp electro-acupuncture or a combination of scalp electro-acupuncture and moxibustion induced a significantly greater reduction in the HAMA and HAMD scores compared to paroxetine. In the study involving patients with GAD [17], the combination of acupuncture and hydro-acupuncture for 4 weeks promoted significantly better total cure rates and marked effective rates compared to 15 mg of buspirone per day, although no significant differences in HAMA, SAS, and TER were found between the groups. In the study [35] involving patients with depression, warming-promotion acupuncture for 1 month promoted significant better total cure rates and HAMD scores compared to regular acupuncture or psychotherapy (Table 1).

**Effectiveness in before–after studies.**   In the two before–after studies [38, 40] involving patients with PTSD, 8 or 12 weeks of acupuncture resulted in a significant improvement in the validated PTSD assessment tool, including CAPS and IES-R-K. Furthermore, these studies [38, 40] showed that acupuncture induced significant improvements in the HAMD, HAMA,

**Table 2. Characteristics of included before–after studies.**

| Study (country) | Sample size (included→analyzed) | Mean age (range) (years) | Sex (M:F) | Population (diagnostic tool) | Related disaster (magnitude/date) | Treatment intervention | Duration of treatment/f/u | Outcome | Results reported | Adverse events reported |
|---|---|---|---|---|---|---|---|---|---|---|
| Wang 2009 (China) [36] | 69 | 31–69 | 26:43 | PTSD (NR) | Wenchuan(8.0/May 2008) | Scalp electro-acupuncture + ear acupressure + moxibustion | 12 weeks | (1) TER | (1) Cure: 38 Improved: 27 Invalid: 4 | NR |
| Yuan 2009 (China) [37] | 34 | 17–62 | 10:24 | PTSD (NR) | Wenchuan(8.0/May 2008) | Acupuncture | 20 days | (1) TER | (1) Cure: 14 Improved: 17 Invalid: 3 | NR |
| Li 2012 (China) [38] | 12 | (40–63) | 1:11 | PTSD (DSM-1V) | Wenchuan earthquake (8.0/May 2008) | Scalp electro-acupuncture | 12 weeks | (1) CAPS (2) HAMD (3) HAMA (4) Treatment compliance* | (1) post-treatment: improved+ (2) post-treatment: improved+ (3) post-treatment: improved+ (4) 10: >80%, 2: >50% | Bleeding or hematoma 4 |
| Moiraghi 2019 (Italy) [39] | 41 | 58.85 ± 13.93 | 8:33 | Patients with musculoskeletal pain and psychologic symptoms (NR) | Central Italy (6.2/August 2016) | Acupuncture or acupuncture + moxibustion | 4 days | Verbal/numerical scale of 1–5** (1) Musculoskeletal pain (2) Psychologic symptoms | (1) post-treatment: improved++ (2) post-treatment: improved++ | Small hematoma 5 |
| Kim 2020 (South Korea) [40] | 16 | 69.38±7.97 | 4:12 | Preliminary PTSD (IES-R-K) | Pohang earthquake (5.5/November 2017) | (1) Ear acupuncture (NADA protocol) | 8 weeks/4 weeks | (1) IES-R-K (2) PHQ-9 (3) PSQI (4–1) STAXI-State (4–2) Trait (4–3) Control (4–4) Out (4–5) In (5–1) EQ-5D-mobility (5–2) Self-care (5–3) Usual activities (5–4) Pain/discomfort (5–5) Anxiety/depression | (1) post-treatment: improved+ (2) post-treatment: improved++, ▲ (3) post-treatment: improved▲ (4–1) post-treatment: N.S (4–2) post-treatment: N.S (4–3) post-treatment: N.S (4–4) post-treatment: N.S (4–5) post-treatment: improved++, ▲ (5–1) post-treatment: improved▲ (5–2) post-treatment: N.S (5–3) post-treatment: N.S (5–4) post-treatment: improved (5–5) post-treatment: N.S | None |

* Treatment compliance = the actual number of treatments received /the number of treatments that should be received * 100%

** 1 = not at all, 2 = slight, 3 = moderate, 4 = severe, 5 = extremely severe

"+" and "++" indicate significant differences between before and after treatment, p < 0.05 and p < 0.01, respectively.

"▲" and "▲▲"indicate significant differences between 4 weeks of treatment and post-treatment; p < 0.05 and p < 0.01, respectively. "N.S" means no significant difference between the two groups, p > 0.05.

Abbreviations: CAPS, Clinician-Administered PTSD Scale; DSM, Diagnostic and Statistical Manual of Mental Disorders; EQ-5D, EuroQol 5-Dimensional; HAMA, Hamilton Anxiety Rating Scale; HAMD, Hamilton Depression Rating Scale; IES-R-K, Impact Event Scale-Revised Korean version; PHQ-9, Patient Health Questionnaire-9; PSQI, Pittsburgh Sleep Quality Index; PTSD, Post-Traumatic Stress Disorder; STAXI, State–Trait Anger Expression Inventory; TER, Total effective rate.

and PHQ-9 scores but not in STAXI scores. A study by Kim et al. (2020) [40], which evaluated the quality of life among participants, found that acupuncture significantly improved some aspects of the EQ-5D, including mobility and pain/discomfort, but not self-care, usual activities, and anxiety/depression. The remaining two before–after studies [36, 37] involving patients with PTSD showed that TERs reached as high as 90%–95% 12 weeks or 20 days after acupuncture. In a study [39] of patients with musculoskeletal pain and psychologic symptoms, 4 days of acupuncture or acupuncture combined with moxibustion significantly improved both musculoskeletal pain and psychologic symptoms (Table 1).

**Safety.**   Five studies [17, 33, 38–40] reported adverse events related to acupuncture, including mild bleeding, hematoma, pain, and syncope. Except for syncope after acupuncture, all other adverse reactions were mild (Table 1).

## Discussion

The current scoping review aimed to identify the current research status of acupuncture for MPD in earthquake survivors based on nine intervention studies. Despite the small number of studies included, the current paper does provide an overview of the potentially promising research status in this field.

### Main findings and interpretation

**Question 1: Which clinical research designs were used in previous investigations on the use of acupuncture to treat MPD?.**   Regarding the study type, some clinical studies used acupuncture to treat MPD following an earthquake, with RCTs (44.44%) and before–after studies (55.56%) accounting for around half of them. The study type of interest in this scoping review was not limited to intervention studies, and despite allowing retrospective studies, including case series, the absolute number of included studies still remained small. Nevertheless, it is encouraging that many of the included clinical studies were RCTs that generate high-quality clinical evidence. Given that RCTs remain the gold standard for effective research [41], this research design should be further encouraged in the field of acupuncture. However, given the difficulty of implementing intervention studies following disasters, such as earthquakes, observational studies, such as retrospective cohort studies or case series, may be useful, highlighting the need for more clinical studies in this field need regardless of study type.

**Question 2: Which populations were targeted in previous MPD acupuncture studies?.** Regarding the population, the most frequent type of MPD observed among the included studies was PTSD (66.67%). However, patients with GAD (11.11%), depression (11.11%), and musculoskeletal pain and psychologic symptoms (11.11%) were also considered target populations. Although PTSD is a common mental health problem after earthquakes, we believe that anxiety and depressive disorders among earthquake survivors may be understudied areas in acupuncture research considering the similarly high prevalence of depression and anxiety [42]. Moreover, given that physical pain commonly occurs after earthquake exposure and is associated with poor long-term physical function [43] and that acupuncture has been found to promote pain relief based on high-quality evidence [44], acupuncture research on earthquake survivors with physical pain and psychological symptoms may also be an attractive area for research.

**Questions 3 & 4: What are the frequently used acupuncture types for MPD management? How long should acupuncture treatment be administered for MPD management?.**   Regarding the acupuncture type, scalp electro-acupuncture were the most frequently used (44.44%), followed by manual acupuncture (33.33%), ear acupressure or ear acupuncture (33.33%), and hydro-acupuncture (11.11%). Interestingly, studies [33, 34, 36, 38] using scalp electro-acupuncture all used common acupoints, including GB20, GV20, GV24, and EX-HN1.

The electrical stimulation method was also mostly similar to continuous wave, with the frequency of 100 Hz. Although studies [17, 35, 37] on manual acupuncture stimulated acupoints distributed throughout the body, EX-HN1 was the most commonly used acupoint of importance. Likewise, common ear acupoints of importance in the studies [33, 36, 40] using ear acupressure or ear acupuncture included the subcortex, sympathetic, liver, and kidney. Except for one before–after study [39] in which 4 days of treatment was performed, the remaining studies [17, 33–38, 40] showed that the treatment period typically lasted between 4 and 12 weeks, and all studies [17, 33–40] reported significant improvements in at least one of the outcomes investigated after acupuncture. Thus far, scalp electro-acupuncture has been reported to have beneficial effects on various neuropsychiatric diseases, including vascular dementia [45], autism [46], stroke [47], post-stroke depression [48], and MDD [49]. Although the underlying therapeutic mechanism by which scalp electro-acupuncture promotes beneficial effects against various neuropsychiatric disorders needs to be further elucidated, this approach has the potential for neuroprotective effects [50, 51] and modulating the functional connectivity between brain regions [52]. According to our findings, EX-HN1 and GV24 were the main acupoints deserving attention in this field. These acupoints have long been considered a major treatment point for psychiatric disorders, with one study showing that electro-acupuncture to these acupoints for 12 weeks significantly improved PTSD symptoms, which is likely associated with enhanced or inhibited functional connectivity between brain regions including the parietal lobe, hippocampus, parahippocampal gyrus, and amygdaloid [53].

**Question 5: Which clinical outcomes were adopted in previous studies on MPD management?.** Regarding outcomes, our findings showed that validated assessment tools for PTSD severity, including CAPS and IES-R-K, as well as assessment tools for accompanying symptoms, including the HAMA, HAMD, PHQ-9 and PSQI, were frequently used for patients with PTSD. However, some of the included studies [36, 37] used only TER as their outcomes without employing a validated evaluation tool for PTSD and/or accompanying symptoms. In general, TER in Chinese clinical studies, which measures the number of participants who responded to treatment, is not considered a disease-specific or validated assessment tool, with possible variations in the TER criteria among studies. Hence, there is room for further improvement in the outcomes of some studies in this field. For patients with other diagnoses or clinical symptoms, corresponding evaluation tools were used. For instance, the HAMA and SAS were used for patients with GAD [17], the HAMD was used for patients with depression [35], and the verbal/numerical scale was used for patients with musculoskeletal pain and psychologic symptoms [39].

**Question 6: What type of adverse events occurred after acupuncture therapy for MPD?.** Regarding the safety of acupuncture, our findings showed that adverse events were poorly reported with none of the studies conducting causality assessment. Nevertheless, considering the reports from five studies [17, 33, 38–40], acupuncture-related adverse events were generally mild and temporary, such as mild bleeding, hematoma, and local pain. Though rarely observed, syncope was the only potentially serious adverse event [17, 33]. Although acupuncture has been considered generally safe [54], future research in this area should still rigorously report the safety profile of the acupuncture procedure considering the presence of factors that increase the risk of acupuncture-related adverse events, such as the risk of invasive infections at disaster sites [55].

## Limitations of this review

One strength of the current scoping review was our ability to, for the first time, provide an overview of the research status on acupuncture for MPD after earthquakes. However, some

limitations should be acknowledged. *First*, given the lack of restrictions on concurrent treatment with acupuncture, outcomes, including adverse events, frequency, or duration of acupuncture, may be obscured. The design of the included studies, for example, when various concurrent treatments were implemented, may affect the interpretation of acupuncture research for MPD after earthquake. *Second*, despite the comprehensive literature search by the authors, potentially related articles in languages other than English, Korean, Japanese, and Chinese may have been excluded. This is because although this review did not limit the language of publication, it was not possible to search local medical databases in all countries using acupuncture. This limitation may potentially explain the insufficient number of clinical studies included in this scoping review. *Third*, consultation, which is the last step of the scoping registration and an optional sixth step, was not conducted in this scoping review due to the difficulty and ethical impropriety associated with artificially setting up patients with mental disabilities. *Fourth*, the methodological quality of the included studies is limited. Importantly, because there were no double-blind studies and assessment blinding was conducted in only two studies [33, 38], the results found in these studies may be exaggerated.

## Conclusion

Acupuncture studies for MPD after earthquake have mainly focused on PTSD. Regarding the type of study, RCTs, which have been considered to provide the highest quality of evidence, accounted for around half of the included studies. Moreover, we found that the scalp electro-acupuncture was commonly used acupuncture type, with EX-HN1 and GV24 being important acupoints in the acupuncture procedures for MPD. Although most of the included studies used validated symptom assessment tools, some did not. Considering our findings, we believe that clinical studies in this field need to be further expanded regardless study type.

## Supporting information

**S1 Appendix. Search terms used in each database.**
(DOCX)

**S2 Appendix. Quality assessment results of the included studies.**
(DOCX)

## Author Contributions

**Conceptualization:** Jungtae Leem, Sang-Ho Kim.

**Data curation:** Da-Woon Kim, Hui-Ju Kwon, Hyun-Seo Park.

**Formal analysis:** Da-Woon Kim.

**Investigation:** Hui-Ju Kwon, Hyun-Seo Park.

**Methodology:** Chan-Young Kwon, Jungtae Leem.

**Project administration:** Sang-Ho Kim.

**Software:** Da-Woon Kim.

**Supervision:** Sang-Ho Kim.

**Writing – original draft:** Chan-Young Kwon, Jungtae Leem.

**Writing – review & editing:** Sang-Ho Kim.

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
