## [Decision Letter · Decision Letter 0]

19 Apr 2023

PONE-D-23-07499Effects of acupuncture on earthquake survivors with major psychiatric disorder and related symptoms: A scoping review of clinical studiesPLOS ONE

Dear,

Thank you for submitting your manuscript to PLOS ONE. After careful consideration, we feel that it has merit but does not fully meet PLOS ONE’s publication criteria as it currently stands. Therefore, we invite you to submit a revised version of the manuscript that addresses the points raised during the review process. It is suggested that the studies of all languages can be included, which will make the results more credible. A bias risk assessment was recommended for all included studies to clarify the evidential value level of the results. 

We look forward to receiving your revised manuscript.

Kind regards,

Muhammad Shahzad Aslam, Ph.D.,M.Phil., Pharm-D

Academic Editor

PLOS ONE

Journal Requirements:

Reviewers' comments:

Reviewer's Responses to Questions

**Comments to the Author**

1. Is the manuscript technically sound, and do the data support the conclusions?

Reviewer #1: Yes

Reviewer #2: Partly

2. Has the statistical analysis been performed appropriately and rigorously? 

Reviewer #1: Yes

Reviewer #2: N/A

3. Have the authors made all data underlying the findings in their manuscript fully available?

Reviewer #1: Yes

Reviewer #2: Yes

4. Is the manuscript presented in an intelligible fashion and written in standard English?

Reviewer #1: Yes

Reviewer #2: No

5. Review Comments to the Author

Reviewer #1: A good and thorough review of available material. The paper is well written and outcomes are described well. I did not find material that needed edits or changes to the current proof of the paper. Well done.

Reviewer #2: This scoping review was for the first time, provide an overview of the research status on acupuncture for MPD after earthquakes. Which is of great research value. As far as I am concerned, there are only a few studies included in this study, which are concentrated after the Wenchuan earthquake. Therefore, it is suggested that the studies of all languages can be included, which will make the results more credible. A bias risk assessment was recommended for all included studies to clarify the evidential value level of the results. It is recommended to include references to individual sources, e.g "The STandards for Reporting Interventions in Clinical Trials of Acupuncture reporting guideline was used for the detailed analysis of the acupuncture regimen used." Further standardize the language of the article, e.g "(4) what was the appropriate length of acupuncture treatment for MPD management?", inaccuracy of expression.

6. PLOS authors have the option to publish the peer review history of their article (what does this mean?). If published, this will include your full peer review and any attached files.

Reviewer #1: **Yes: **Jennifer E Brett

Reviewer #2: No

---

## [Author Response · Author response to Decision Letter 0]

2 May 2023

Reviewer #1: A good and thorough review of available material. The paper is well written and outcomes are described well. I did not find material that needed edits or changes to the current proof of the paper. Well done.

Thank you for your dedication to reviewing this manuscript.

Reviewer #2: This scoping review was for the first time, provide an overview of the research status on acupuncture for MPD after earthquakes. Which is of great research value.

Thank you for your dedication to reviewing this manuscript. We have improved the quality of this manuscript by reflecting your comments in this revised manuscript. All corrections in the manuscript are described with red words.

As far as I am concerned, there are only a few studies included in this study, which are concentrated after the Wenchuan earthquake. Therefore, it is suggested that the studies of all languages can be included, which will make the results more credible.

Thank you for the comments. We think that this is a misunderstanding that occurred because the inclusion criteria in our original manuscript were not clearly described. That is, this review did not place any restrictions on the language of publications. Nevertheless, the reason for the concentration of research in some countries appears to be related to the predominant use of acupuncture in Eastern cultures. We have added the following sentences to clarify this point.

Line 153:

“However, case reports with fewer than three patients (28), literature reviews, and preclinical studies were excluded. There were no restrictions on the publication language of the study.”

Lines 462-465:

“Second, despite the comprehensive literature search by the authors, potentially related articles in languages other than English, Korean, Japanese, and Chinese may have been excluded. This is because although this review did not limit the language of publication, it was not possible to search local medical databases of all countries using acupuncture.”

A bias risk assessment was recommended for all included studies to clarify the evidential value level of the results.

Thank you for the comments. We have added an assessment of the methodological quality of the included studies in this revised manuscript.

Lines 201-208:

“Also, the methodological quality of the studies included in this review was assessed. Corresponding assessment tools were used to assess methodological quality depending on the type of study included. Since the study design of the included studies in this review were RCTs and before-after studies, the Cochrane’s risk of bias tool (31) and Quality Assessment Tool for Before-After (Pre-Post) Studies With No Control Group developed by the National Heart, Lung, and Blood Institute (URL: https://www.nhlbi.nih.gov/health-topics/study-quality-assessment-tools) were used, respectively. The methodological quality of the included studies was conducted by two independent researchers (DWK and SHK), and in case of a disagreement in the evaluation process, the disagreement was resolved by their discussion.”

Lines 335-345:

“Methodological qualities of included studies

Among the four RCTs (17, 32, 33, 34) included, two (17, 32) were considered to use random sequence generation with a low risk of bias. Only one study (32) performed and described allocation concealment. All studies (17, 32, 33, 34) were rated as unfavorable in terms of double-blindness. There was only one study (32) reporting blinding of outcome assessment. All five included before-after studies (35, 36, 37, 38, 39) clearly stated the purpose of the study. However, only one study (37) clearly stated the criteria for inclusion and exclusion of participants. Except for one study (38), the interventions used were clearly described in the remaining studies. Two studies (37, 39) clearly described the use of the outcome used. Only one study (37) reported blinding of outcome assessment. Follow-up was conducted in one study (39), but the loss to follow-up was more than 20%. In three studies (37, 38, 39), appropriate statistical analysis was performed for before-and-after comparisons, p-values were presented, and multiples times outcome indicators were evaluated after intervention (S2 Appendix).”

It is recommended to include references to individual sources, e.g "The STandards for Reporting Interventions in Clinical Trials of Acupuncture reporting guideline was used for the detailed analysis of the acupuncture regimen used."

Thank you for the comment. In this revised manuscript, we have cited proper references that were insufficiently cited.

Line 116:

“… the Preferred Reporting Items for Systematic reviews and Meta-Analyses Extension for Scoping Reviews guidelines (26).”

(26) Tricco AC, Lillie E, Zarin W, O'Brien KK, Colquhoun H, Levac D, et al. PRISMA extension for scoping reviews (PRISMA-ScR): checklist and explanation. Annals of internal medicine. 2018;169(7):467-73.

Line 197:

“The STandards for Reporting Interventions in Clinical Trials of Acupuncture reporting guideline (30) was used …”

(30) MacPherson H, Altman DG, Hammerschlag R, Youping L, Taixiang W, White A, et al. Revised STandards for Reporting Interventions in Clinical Trials of Acupuncture (STRICTA): Extending the CONSORT statement. J Evid Based Med. 2010;3(3):140-55.

Further standardize the language of the article, e.g "(4) what was the appropriate length of acupuncture treatment for MPD management?", inaccuracy of expression.

Thank you for the comment. After making efforts to revise this manuscript based on the above comments, we requested a review from an English proofreading company and received professional academic English review and proofreading services.

---

## [Decision Letter · Decision Letter 1]

22 May 2023

Effects of acupuncture on earthquake survivors with major psychiatric disorders and related symptoms: A scoping review of clinical studies

PONE-D-23-07499R1

Dear,

We’re pleased to inform you that your manuscript has been judged scientifically suitable for publication and will be formally accepted for publication once it meets all outstanding technical requirements.

Kind regards,

Muhammad Shahzad Aslam, Ph.D.,M.Phil., Pharm-D

Academic Editor

PLOS ONE

Additional Editor Comments (optional):

Reviewers' comments:

Reviewer's Responses to Questions

**Comments to the Author**

1. If the authors have adequately addressed your comments raised in a previous round of review and you feel that this manuscript is now acceptable for publication, you may indicate that here to bypass the “Comments to the Author” section, enter your conflict of interest statement in the “Confidential to Editor” section, and submit your "Accept" recommendation.

Reviewer #2: All comments have been addressed

2. Is the manuscript technically sound, and do the data support the conclusions?

Reviewer #2: (No Response)

3. Has the statistical analysis been performed appropriately and rigorously? 

Reviewer #2: (No Response)

4. Have the authors made all data underlying the findings in their manuscript fully available?

Reviewer #2: (No Response)

5. Is the manuscript presented in an intelligible fashion and written in standard English?

Reviewer #2: (No Response)

6. Review Comments to the Author

Reviewer #2: (No Response)

7. PLOS authors have the option to publish the peer review history of their article (what does this mean?). If published, this will include your full peer review and any attached files.

Reviewer #2: No

---

## [Editor Report · Acceptance letter]

31 May 2023

PONE-D-23-07499R1 

Effects of acupuncture on earthquake survivors with major psychiatric disorders and related symptoms: A scoping review of clinical studies 

Dear Dr. Kim:

I'm pleased to inform you that your manuscript has been deemed suitable for publication in PLOS ONE. Congratulations! Your manuscript is now with our production department. 

Kind regards, 

on behalf of

Dr. Muhammad Shahzad Aslam 

Academic Editor

PLOS ONE